# A robust fuzzy logic-based model for predicting the critical total drawdown in sand production in oil and gas wells

**Fahd Saeed Alakbari**[1], **Mysara Eissa Mohyaldinn**[1]\*, **Mohammed Abdalla Ayoub**[1], **Ali Samer Muhsan**[2], **Ibnelwaleed A. Hussein**[3]

**1** Petroleum Engineering Department, Universiti Teknologi PETRONAS, Bandar Seri Iskandar, Perak, Malaysia, **2** Mechanical Engineering Department, Universiti Teknologi PETRONAS, Bandar Seri Iskandar, Perak, Malaysia, **3** Gas Processing Center, College of Engineering, Qatar University, Doha, Qatar

\* mysara.eissa@utp.edu.my

## Abstract

Sand management is essential for enhancing the production in oil and gas reservoirs. The critical total drawdown (CTD) is used as a reliable indicator of the onset of sand production; hence, its accurate prediction is very important. There are many published CTD prediction correlations in literature. However, the accuracy of most of these models is questionable. Therefore, further improvement in CTD prediction is needed for more effective and successful sand control. This article presents a robust and accurate fuzzy logic (FL) model for predicting the CTD. Literature on 23 wells of the North Adriatic Sea was used to develop the model. The used data were split into 70% training sets and 30% testing sets. Trend analysis was conducted to verify that the developed model follows the correct physical behavior trends of the input parameters. Some statistical analyses were performed to check the model's reliability and accuracy as compared to the published correlations. The results demonstrated that the proposed FL model substantially outperforms the current published correlations and shows higher prediction accuracy. These results were verified using the highest correlation coefficient, the lowest average absolute percent relative error (AAPRE), the lowest maximum error (max. AAPRE), the lowest standard deviation (SD), and the lowest root mean square error (RMSE). Results showed that the lowest AAPRE is 8.6%, whereas the highest correlation coefficient is 0.9947. These values of AAPRE (<10%) indicate that the FL model could predicts the CTD more accurately than other published models (>20% AAPRE). Moreover, further analysis indicated the robustness of the FL model, because it follows the trends of all physical parameters affecting the CTD.

## Introduction

Approximately 70% of the world's petroleum wells are situated in weakly consolidated reservoirs [1]. Consequently, many petroleum wells are susceptible to sand production that causes several problems, such as equipment damage and plugging, maintenance costs, and decline in reservoir recovery. Moreover, sand production can also lead to equipment erosion and safety,

**Data Availability Statement:** All relevant data are within the paper.

**Funding:** The authors would like to express their appreciation to the Universiti Teknologi PETRONAS

for supporting this work under YUTP-Grant cost centre 15LC0-098. The first author, in particular, is grateful to Universiti Teknologi PETRONAS for supporting his PhD study under Graduate Assistance (GA) scheme.

**Competing interests:** The authors have declared that no competing interests exist.

environmental, and health issues [2–5]. However, an early sand production prediction is highly recommended for the success of sand control management strategies [6].

There are three techniques used to predict sand production: numerical, analytical, and empirical methods. The numerical method includes the application of the finite element method, discrete element method, or finite difference method. However, the numerical methods are time-consuming and involve a complicated process. Furthermore, the input data required to obtain a numerical prediction (such as rock petrophysics, rock mechanics, and fluid properties) are challenging and laborious to find because of the need of some experimental data [7]. Similarly, the analytical methods have some drawbacks; they ignore stress anisotropy and assume symmetrical geometry and boundary conditions. Therefore, ignoring the main effect of stress anisotropy on sand, the method may not explain the sanding risk related to the orientation of the borehole. In general, assumptions or approximations are needed, making the models less reliable or accurate, even though they are complex [7].

On the other hand, empirical methods use well data and field observation to predict sand production. Sand prediction methods depend on field experiences to establish a correlation between sand production, well data, and field and operation parameters. In general, empirical methods are categorized into three types: one parameter, two parameters, and multiparameter correlations [6]. Tixier et al. [8] used acoustic log data to determine the shear modulus ratio to compressibility to obtain sand production. When the ratio is higher than $0.8 \times 10^{12} psi^2$, there is a lower probability of sand influx; and when the ratio is less than $0.7 \times 10^{12} psi^2$, there is a high probability of sand production [8]. Veeken et al. [9] applied a model with two parameters, which are the depletion reservoir pressure and drawdown pressure, as indicators for sand risk. Generally, increasing the number of parameters improves the accuracy of the sand prediction model [9].

Some models and correlations are available in the literature for the prediction of the critical total drawdown (CTD) that is used as an indicator of the onset of sand production. The CTD can be defined as the maximum difference between the reservoir pressure and the minimum well bottomhole flowing pressure that the formation can withstand without sand production. Some researchers used analytical models like Mohr Coulomb and modified Lade to predict the CTD; nevertheless, the models have some assumptions such as the formation rock mechanics properties are homogenous and isotropic [10–12]. Kanj and Abousleiman [13] used ANNs, feed-forward backpropagation network (BPN), and generalized regression neural network to predict the CTD using data of 31 wells from the Adriatic Sea. Multiple linear regression (MLR) and the genetic algorithm MLP (GA-MLR) were applied by Khamehchi et al. [6] to predict the CTD using data of 23 wells from the Adriatic Sea. However, these models are proven to have a lack of accuracy that reaches more than 20% error (AAPRE).

Numerous studies have used the fuzzy logic (FL) approach in petroleum engineering. Rezaee et al. [14] used petrophysical data and applied the FL tool to calculate shear wave velocity, which showed accurate predictions. In addition, Moradi et al. [15] used a FL approach to obtain the drilling rate. The FL model is proved to be more accurate than other models, such as Bourgoyne and Young models [15]. A FL tool was also developed to assist in selecting candidate wells for hydraulic fracturing treatment in a carbonate reservoir [16]. The FL model reduced the uncertainty that existed in the candidate well selection [16]. Ahmadi et al. [17] used the FL to calculate the breakthrough time of water coning in the fractured reservoirs. Akbarzadeh et al. [18] used a fuzzy model to predict conductivity; the fuzzy model was reported to be robust and accurate. Wang et al. [19] applied the FL to characterize reservoir heterogeneity and demonstrated that the model was accurate. The FL model was also used for forecasting petroleum economic parameters; the authors concluded that Mamdani type outperformed other models, such as autoregressive integrated moving average (ARIMA) [20]. Al-

Jamimi and Saleh [21] used an FL tool to optimize the catalysts, and the FL model was successful in predicting catalyst performance. Artun and Kulga [22] used the FL to select candidate wells for refracturing in tight gas sand reservoirs. Karacan [23] used the FL model using 24 data points to estimate the recovery factors of miscible $CO_2$. API, porosity, permeability, depth, hydrocarbon pore volumes (HCPV), net pay, initial pressure, well spacing, and $S_{orw}$ were included as features; the FL model was showed to be accurate [23].

This study aims to build a new robust and more accurate model for predicting the CTD by applying the FL. The developed model considers four parameters: total vertical depth (TVD), transit time (TT), cohesive strength (COH), and effective overburden vertical stress (EOVS). A trend analysis has been performed to investigate the accuracy of the physical behavior and trends of the model parameters. Furthermore, the performance of the model was compared with the most recent correlations.

## Methodology

### Data collection and description

This study has been performed in four phases: data collection and preparation, model development, trend analysis, and validation. A data set of 23 wells of the North Adriatic Sea was collected from the literature [24]. The data were split into two sections: for the first section, 70% of the data sets were allocated for developing the model, and for the second section, 30% of the data were used for verifying the model. Table 1 lists the data range and statistical analysis of the training and verification parameters.

### Fuzzy logic approach

Zadeh [25] invented a fuzzy set theory to handle data uncertainty. The benefit of using FL is that it considers the identification uncertainty present in any evaluation process in the developed model [26]. The FL can deal between zero and one, unlike the Boolean that can only take a zero or one. The fuzzy sets can provide gradual transitions from membership to non-membership [27]. The proposed fuzzy logic model offered high robust and reliable estimations and is thus well suited to other applications. The FL system is flexible and has a structure that can be modified. The FL methods can link human reasoning and concept formation through linguistic rules to obtain functions and control nonlinear systems. The FL can efficiently handle the complexity and uncertainty of the process with limited data [28]. The fuzzy logic can be applied with small data; hence they cannot occupy a huge memory space [29]. The fuzzy inference system contains five functional components, as illustrated in Fig 1:

**Table 1. Data range and statistical analysis of the collected data for the developing FL model.**

| Parameter | TVD (m) | TT (micsec/ft) | COH (Mpa) | EOVS (Mpa) | CTD measured (Mpa) |
|---|---|---|---|---|---|
| Minimum | 1070.000 | 85.000 | 0.539 | 10.885 | 0.314 |
| Maximum | 4548.000 | 170.000 | 5.217 | 80.709 | 43.973 |
| Mean | 2564.957 | 115.043 | 1.775 | 38.165 | 15.284 |
| Median | 2380.000 | 110.000 | 1.275 | 29.420 | 12.807 |
| Range | 3478.000 | 85.000 | 4.678 | 69.823 | 43.659 |
| Skewness | 0.187 | 0.940 | 1.234 | 0.398 | 0.600 |
| Standard deviation | 10.238 | 0.208 | 0.012 | 0.228 | 0.123 |

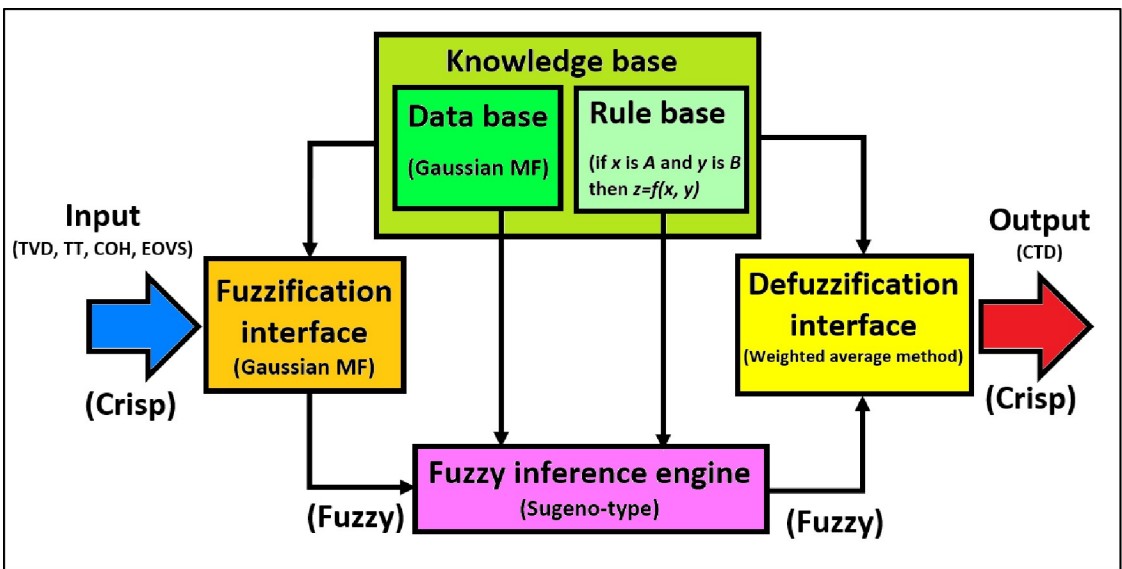

**Fig 1. Fuzzy inference system for CTD model.**

a. A fuzzification interface is used to convert crisp inputs to linguistic (fuzzy) variables applying membership functions.

b. A database is used to define membership functions.

c. A rule base contains several fuzzy if-then rules.

The IF of the rule describes a condition or assumption that is partly satisfied, whereas the THEN of the rule describes a conclusion or an action obtained when the conditions are hold true [26].

d. The inference engine of the FL is a decision-making unit.

e. The defuzzification interface is used to convert fuzzy outputs into crisp outputs [30]. The defuzzification can be conducted using some defuzzification methods such as the max or mean-max membership principles, the centroid method, and the weighted average method [31]. We used the weighted average method for this research.

The FL model in this study was developed by applying MATLAB R2020a. A membership function (MF) is known as the curve, which defines how each point in the input space can be designed to a membership value between (0–1) [32]. An MF can identify the fuzzy set by assigning a membership degree to each element [23]. Fuzziness is measured by using MFs as the fundamental constituents of the fuzzy set theory. The shape and type of MF should be accurately chosen because they impact the fuzzy inference system. Trapezoidal MFs were applied for the independent data because they show significantly enhanced outcomes compared to other MFs, whereas Gaussian MFs were used for the dependent data because they can be non-zero and smooth [33]. Gaussian MFs was used for this study. Table 2 presents the specifications of the FL MATLAB code used to obtain the CTD model.

## Results and discussion

Two tests were performed to evaluate the proposed FL model. First, the FL approach was tested to show that it is robust and follows physical behavior trends by applying trend analysis.

**Table 2. Specifications of the FL model.**

| Parameter | Description/value |
|---|---|
| **Fuzzy structure** | Sugeno-type |
| **Initial FIS for training** | genfis3 |
| **Membership function type** | Gaussian MF |
| **Output membership function** | linear |
| **The number of membership functions** | 4 |
| **The fuzzy rules** | if $x$ is $A$ and $y$ is $B$ then $z = f(x, y)$ |
| **The fuzzification** | Gaussian |
| **The defuzzification** | weighted average method |
| **The number of clusters** | 4 |
| **Number of inputs** | 4 |
| **Number of outputs** | 1 |
| **Training epoch number** | 500 |
| **Radii** | 1.1585 |

Second, the performance of the proposed FL approach was compared with the current correlations. Cross-plots and statistical error analyses, such as correlation coefficient (R), average percent relative error (APRE), average absolute percentage relative error (AAPRE), root mean square error (RMSE), and standard deviation (SD), were conducted.

## Trend analysis

The trend analysis was performed to test the robustness of the model in the presence of uncertainty. The trend analysis is used to indicate the relationships between input and output variables in the model. The trend analysis defines errors in the models to show unexpected relationships between input and outputs, which highlights the need to display the reliability of the models. Furthermore, the trend analysis identifies and removes the unnecessary parts of the model structure [34]. Moreover, the trend analysis was used to identify significant connections among observations, model inputs, and predictions, guiding the development of robust models [35]. Therefore, the trend analysis is essential for this study.

The selected input parameter for investigation is varied between the minimum and maximum value, while other parameters are kept constant at their mean values [36–38]. Graphs are plotted for the input parameter values (x-axis) against the output CTD (y-axis) for the previous models and the FL model. Four input parameters, TVD, TT, COH, and EOVS, have been selected for the trend analysis.

Fig 2 shows the trend of TVD. Kanj and Abousleiman [13] correlation (Fig 2) shows that the CTD was independent of TVD, because it is based only on the cohesive strength (COH). The TVD trend of the FL model obeys the trend of the existing correlations, as shown in Fig 3. Ahad et al. [39] stated that older rocks can be more consolidated. On the other hand, shallow formations can be weakly consolidated [39]. Therefore, increasing the depth will increase the CTD.

Fig 4 indicates that the TT is inversely proportional to the CTD, as illustrated by all the previous models except Kanj and Abousleiman [13] correlation, which demonstrates that the CTD is constant as they did not include the TT. As a result, Kanj and Abousleiman [13] correlation failed to represent the behavior accurately. The FL model also follows the trend of the existing correlations (Fig 5), indicating the proper trend for the TT. The shorter TT implies that the sand is more consolidated [40]. Consequently, decreasing TT will increase the CTD.

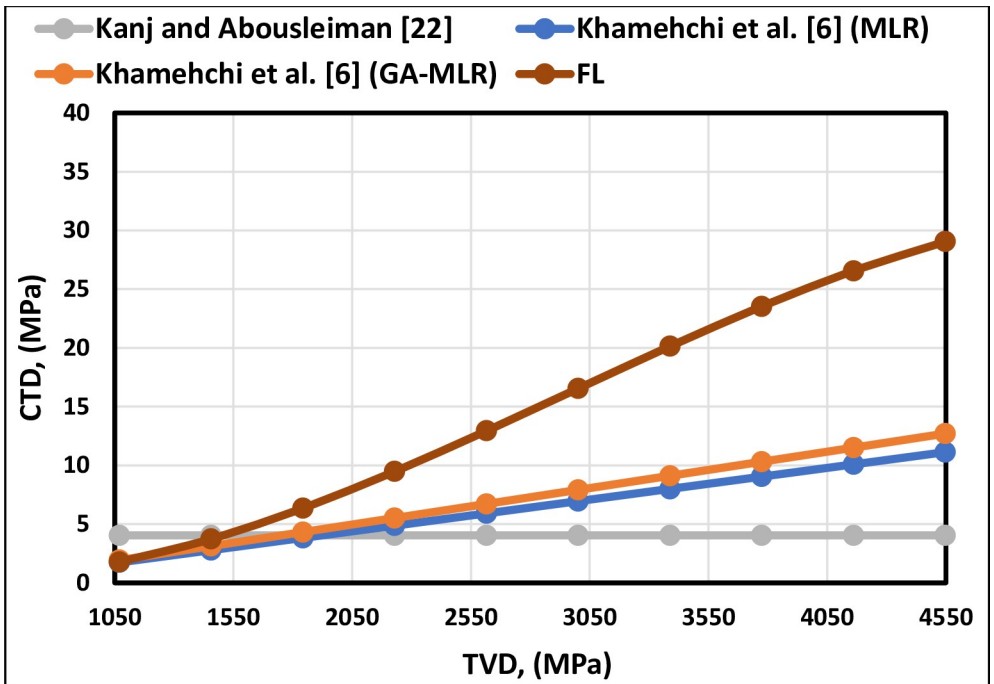

**Fig 2. TVD trend analysis of the FL model and previously published models.**

Fig 6 indicates that the cohesive strength (COH) is directly proportional to the CTD. Kanj and Abousleiman [13] correlation followed the trend of existing correlations, but the CTD is negative (−2.57 MPa) when the COH is 0.539 MPa. Consequently, Kanj and Abousleiman [13] correlation has not proven a proper trend for the CTD correlation (Fig 6). The FL model shows that the COH trendobeys the trend shown by the correlations in the literature where the COH is directly proportional to the CTD (Fig 7). Hence, the FL model is successful in

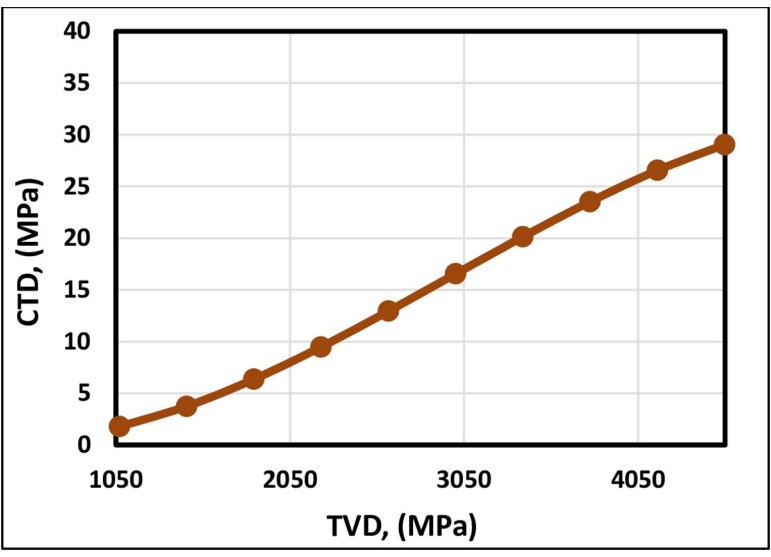

**Fig 3. TVD trend analysis of the FL model.**

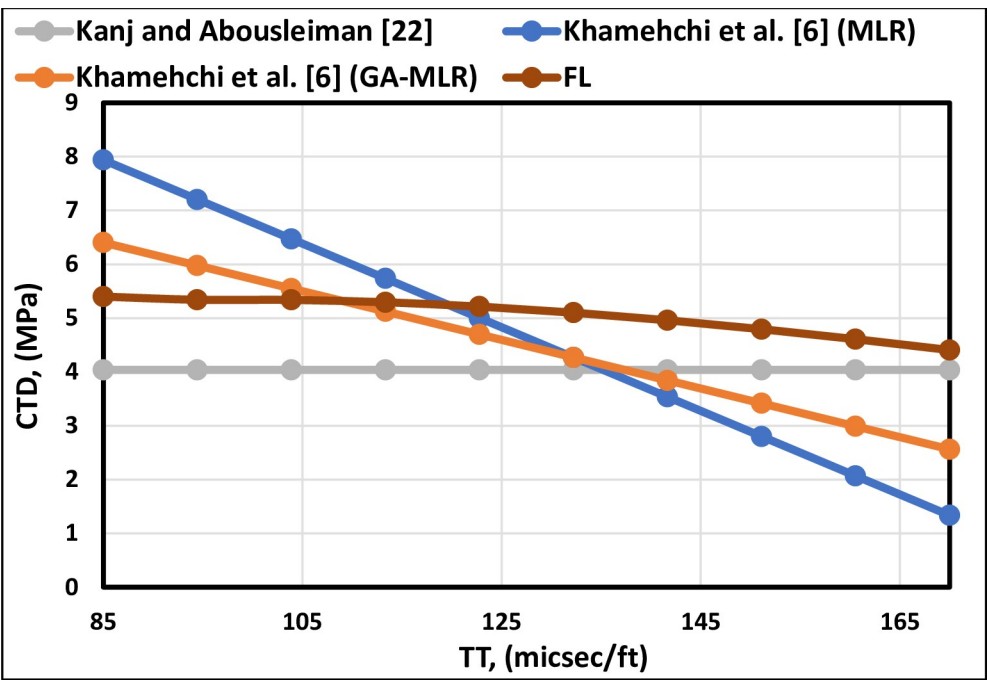

**Fig 4. TT trend analysis of the FL model and previously published models.**

following the accurate trends. The formation failure decreases the strength of rock and causes sand production [41]. The cohesive strength increases the degree of cementation [42]. Increasing the cementation degree of sand grains can lead to a decrease in sand production. Thus, increasing the rock's cohesive strength results in in rising the CTD.

The trend of the EOVS is illustrated in Fig 8. The CTD follows an inverse relationship with EOVS. However, Kanj and Abousleiman [13] correlation displays a horizontal line, which indicates that their correlation does not consider the EOVS parameter. The trend expressed by

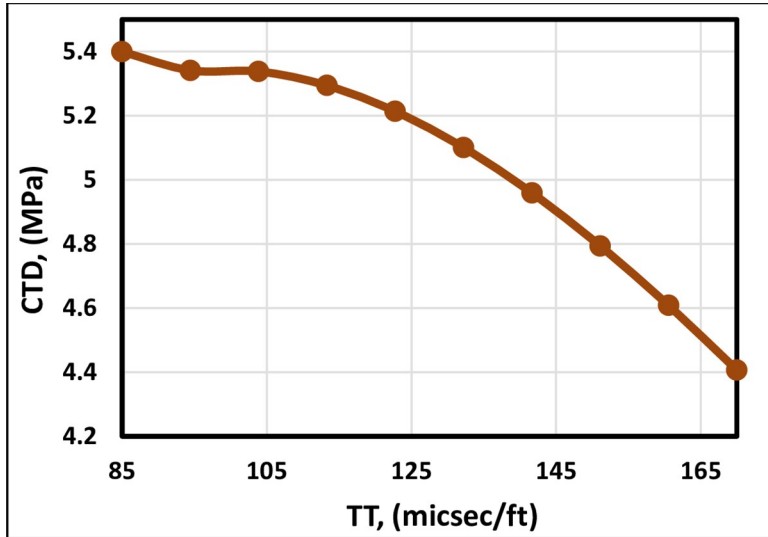

**Fig 5. TT trend analysis of the proposed FL model.**

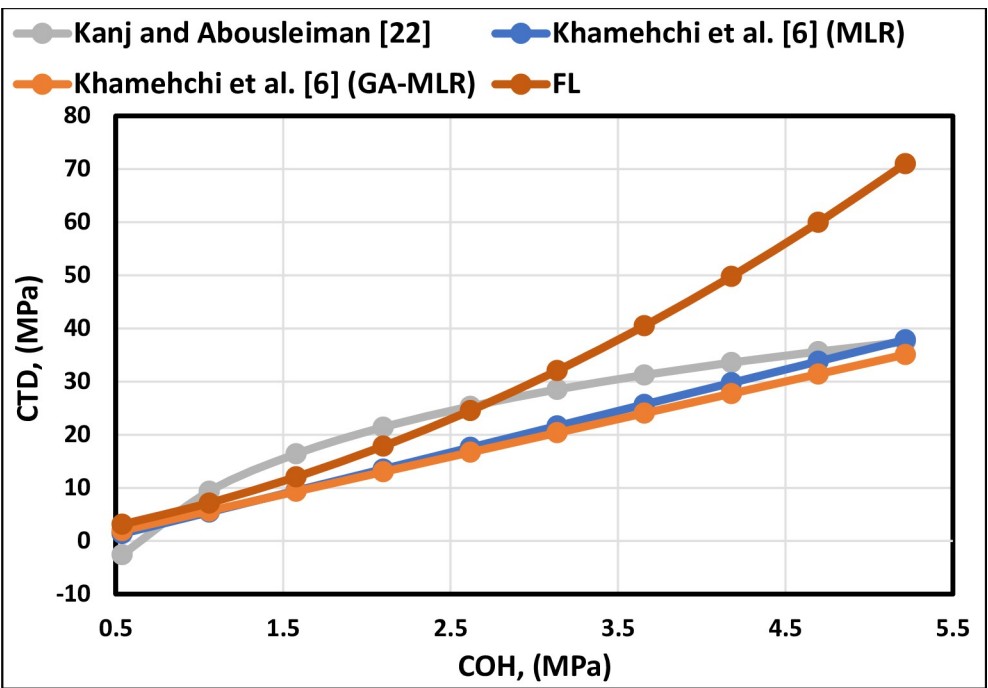

**Fig 6. COH trend analysis of the proposed FL model and previously published models.**

the FL model is also shown to follow the trend of the previous correlations; Fig 9 indicates that it represents the proper trend for the EOVS. The overburden stress stays constant; however, when the pore pressure declines, the effective overburden stress must rise [42]. The critical drawdown pressure decreases with the decline in pore pressure [43]. Therefore, increasing EOVS decreases the CTD.

To summarize the trend analysis, all the input parameters (TVD, TT, COH, and EOVS) of the developed FL model can follow the correct trends, indicating the FL model's reliability.

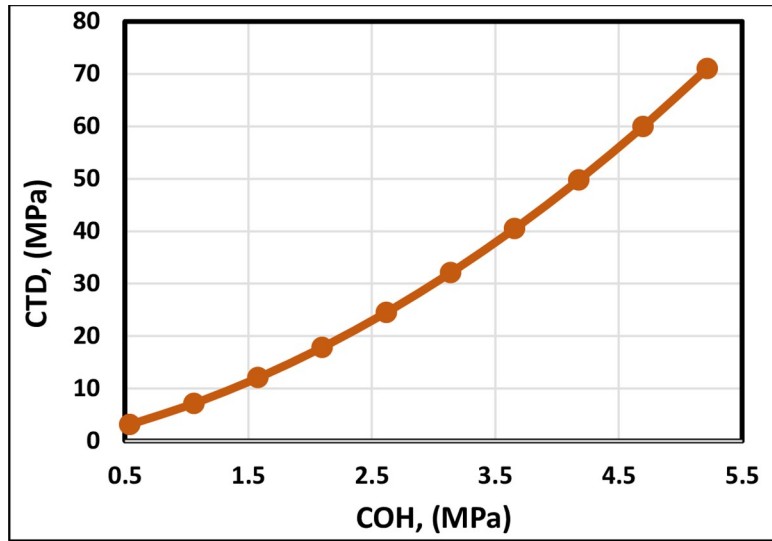

**Fig 7. COH trend analysis of the FL model.**

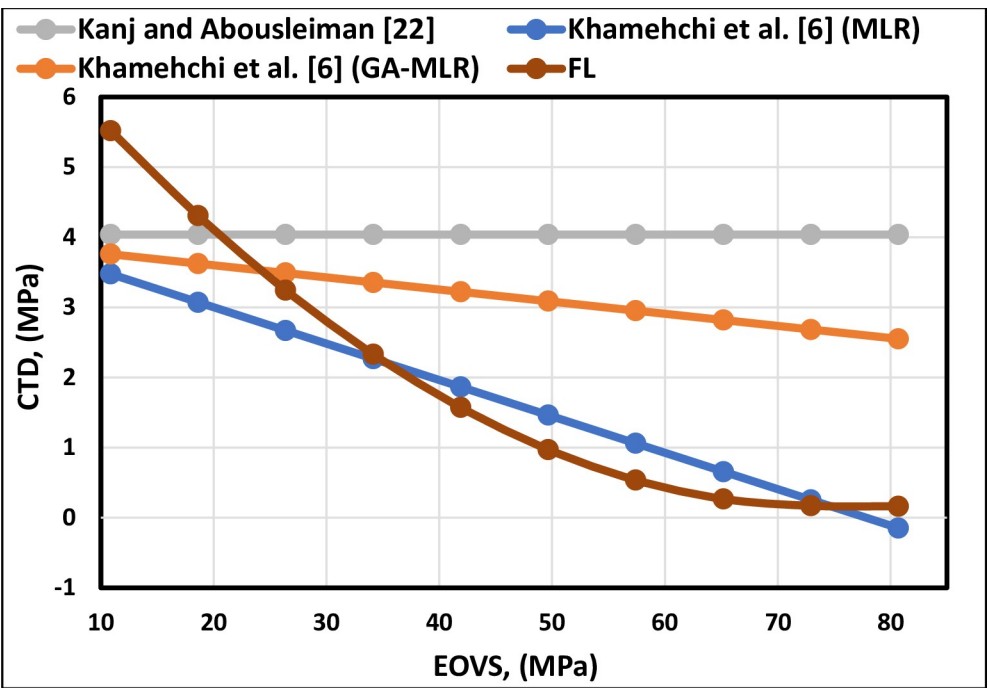

**Fig 8. EOVS trend analysis of the FL model and previously published models.**

Nevertheless, Kanj and Abousleiman [13] correlation trends fail to present the behavior correctly.

## The comparison of FL model and current models

**Cross-plotting analysis.** The proposed FL model and current correlation cross-plots were presented. A 45˚ straight line is illustrated on the cross-plot of the measured and expected

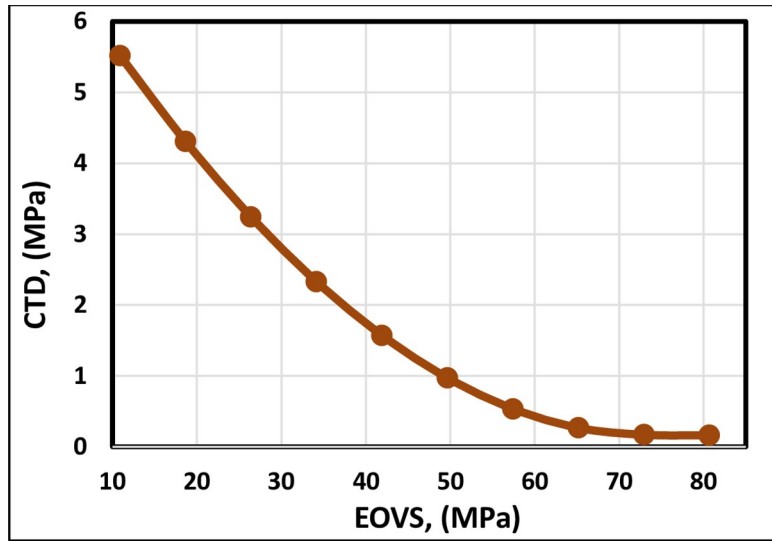

**Fig 9. EOVS trend analysis of the proposed FL model.**

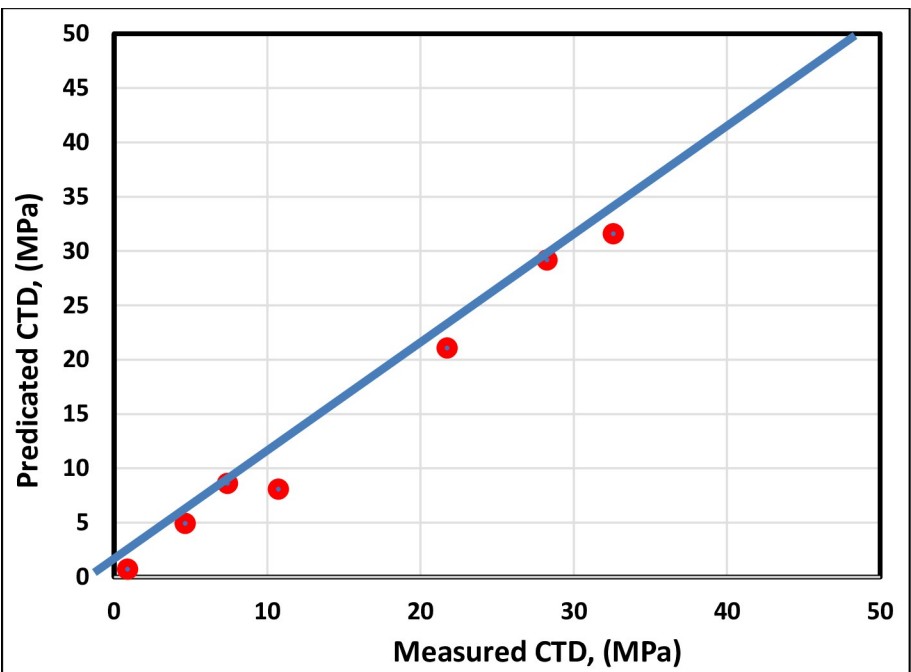

**Fig 10. Cross-plot of testing FL model.**

CTD values. The closer the plotted data points to the straight line, the higher the correlation or model's accuracy.

Fig 10 illustrates the cross-plotting of the testing data set of the FL model, and Fig 11 illustrates the cross-plotting comparison of the FL model with the existing correlations. As shown

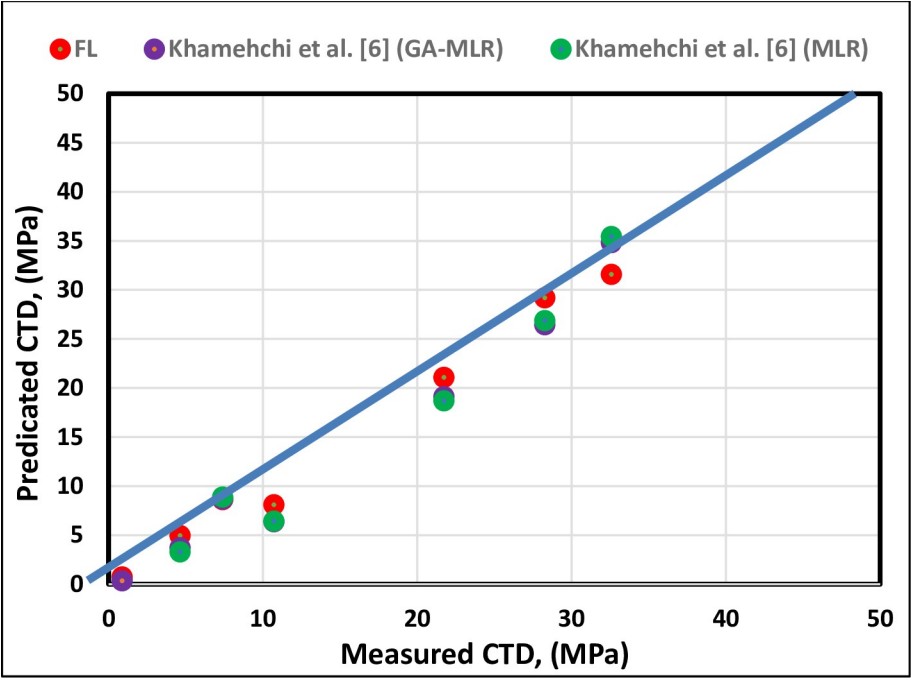

**Fig 11. Cross-plot comparison of the proposed FL model with the previously published models.**

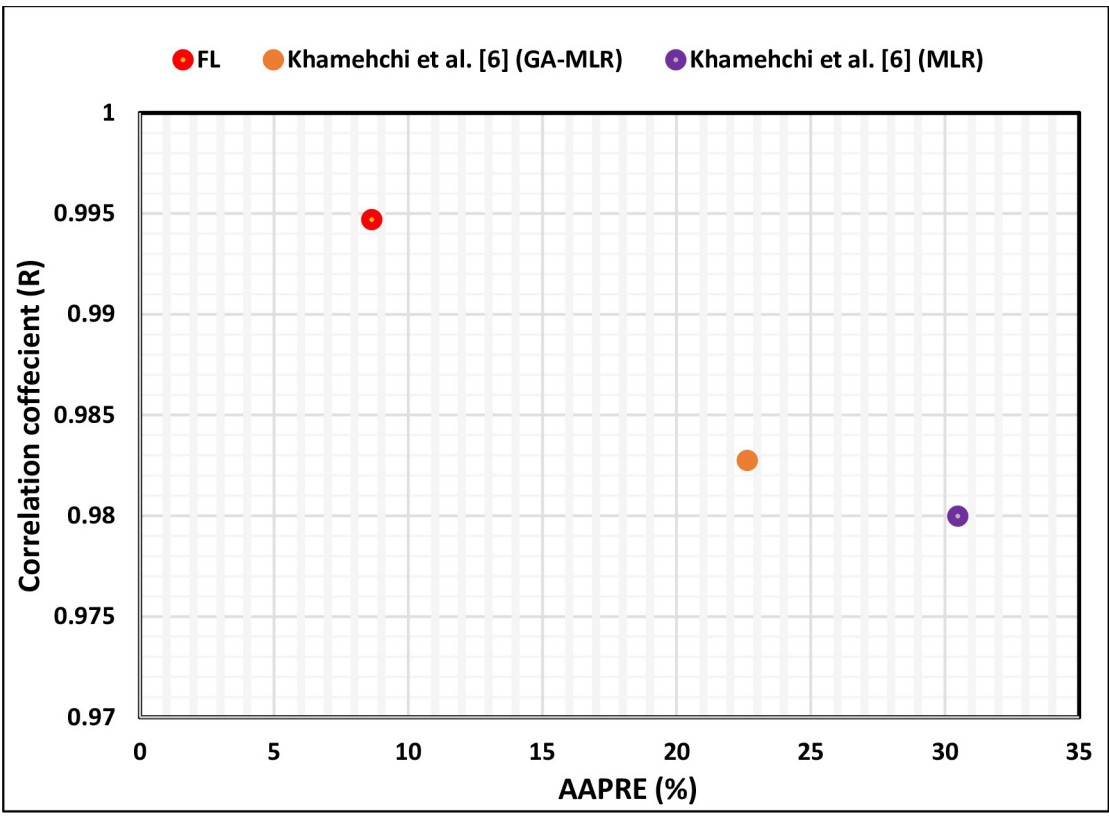

**Fig 12. Correlation coefficient (R) and AAPRE (%) comparison of the proposed FL model with the previously published models.**

in Figs 10 and 11, the FL model represents the highest accuracy and can predict the CTD with the coefficient determination ($R^2$) of 0.9947.

**Statistical error analysis.** The statistical error analysis is performed to verify the FL model's accuracy and compare it against the current correlations. The statistical parameters used in this research are correlation coefficient (R), APRE, AAPRE, SD, RMSE, maximum absolute percent relative error ($E_{max.}$), and the minimum absolute percent relative error ($E_{min.}$), as included in the S1 Appendix. The AAPRE and correlation coefficient (R) are used as indicators.

Fig 12 illustrates the AAPRE and correlation coefficient (R) comparison of the FL model with the existing models. As shown in Fig 12, the proposed FL model has the lowest AAPRE of 8.647% and the highest correlation coefficient (R) of 0.9947. Khamehchi et al. [6] (GA-MLR) model shows the AAPRE (%) of 22.644% and correlation coefficient (R) of 0.9827, whereas Khamehchi et al. [6] (MLR) model shows the highest AAPRE of 30.485%.

The published predictions of the performance correlations were compared with the proposed FL approach, as shown in Fig 13. Statistical error analysis has been conducted to test the robustness of the proposed FL model. The FL model also has the lowest RMSE and SD compared to other models (Fig 13). This comparison of all correlations and the FL model provides essential means for validating the performance of the proposed FL model. Investigation of these statistical error analyses indicates that the FL model outperforms all the existing correlations. The AAPRE and correlation coefficient (R) are taken as the primary indicators of accuracy in this study. Khamehchi et al. [6] (GA-MLR) correlation is ranked as the second

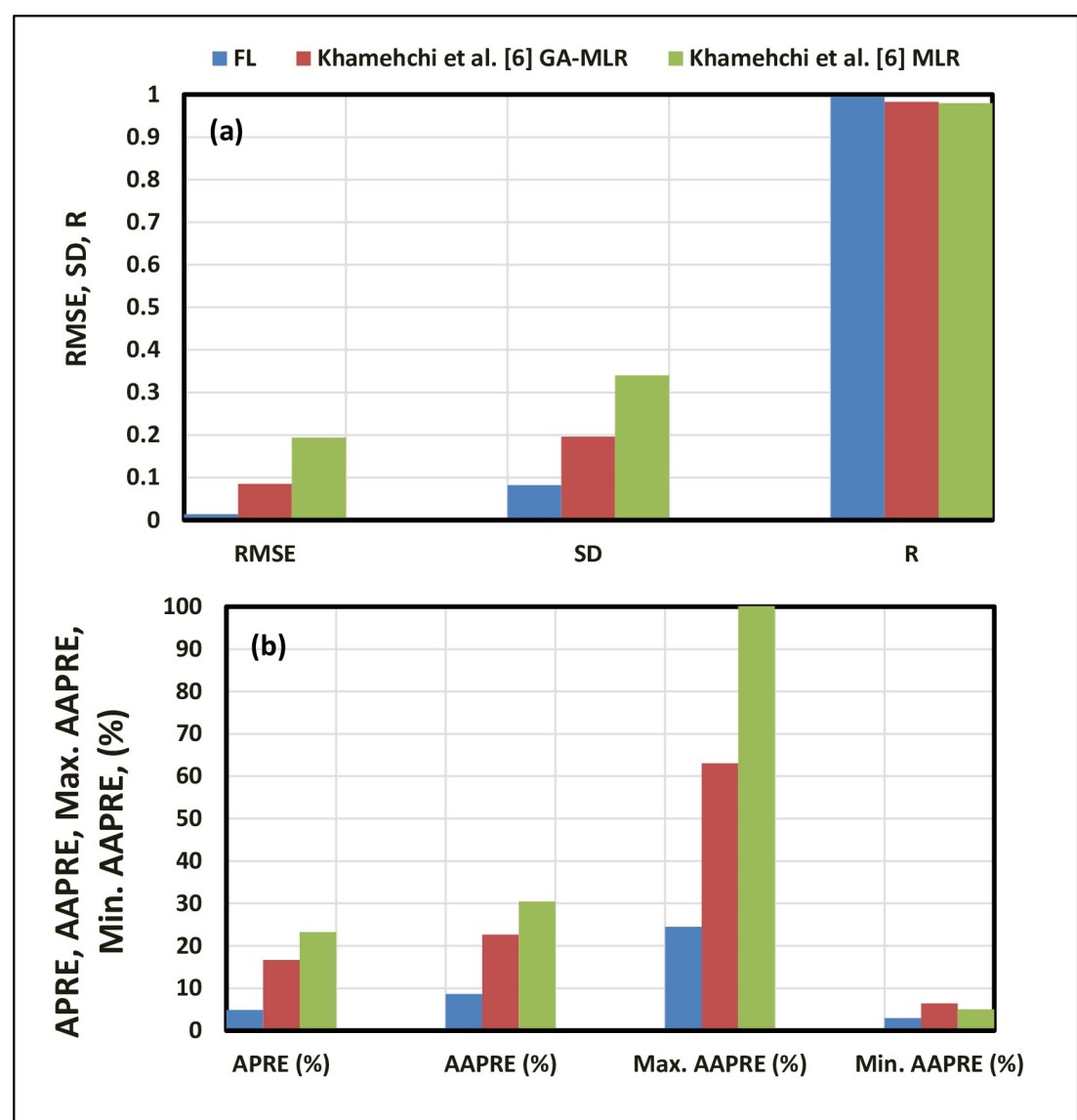

**Fig 13. Comparison of the statistical parameters for the proposed FL model with other models.** (a) RMSE, SD, and R; (b) APRE, AAPRE, $AAPRE_{max}$, and $AAPRE_{min}$.

correlation; it has an AAPRE of 22.644% and a correlation coefficient (R) of 0.9827. Khamehchi et al. [6] (GA-MLR) and (MLR) models show AAPRE of more than 20% (Fig 13).

## Conclusions

An FL model was developed for predicting the CTD in sand formations in oil and gas wells. Different techniques, such as trend analysis, cross-plotting, and statistical error analysis, were used to validate the model. The prediction outcomes were compared with the published models available in the literature. Based on the obtained results, the following conclusions are emphasized:

• The FL model could accurately describe the proper trend of the CTD as a function of all the considered independent variables (i.e., TVD, TT, COH, and EOVS). The model is observed to follow the actual trend as expected from the physical relationship.

- The FL model has provided the best CTD estimations as compared to other available correlations. The FL model presented the highest correlation coefficient of 0.9947, the lowest AAPRE of 8.647%, the lowest root mean squared error of 0.014, and the lowest SD of 0.082 compared to the published correlations. The model accuracy and reliability have further been enhanced by data randomization to ensure that each data set does not memorize the pattern and avoid generalization and model overfitting.

- The FL model has shown an AAPRE of 8.647%, whereas the existing models have reported values higher than 20%.

## Supporting information

**S1 Appendix.**
(DOCX)

## Acknowledgments

Special thanks to the Centre of Research in Enhanced Oil Recovery (COREOR), Petroleum Engineering, Universiti Teknologi PETRONAS for supporting this work.

## Author Contributions

**Conceptualization:** Mysara Eissa Mohyaldinn.

**Data curation:** Fahd Saeed Alakbari.

**Funding acquisition:** Mysara Eissa Mohyaldinn.

**Methodology:** Fahd Saeed Alakbari.

**Software:** Fahd Saeed Alakbari.

**Supervision:** Mysara Eissa Mohyaldinn.

**Validation:** Mohammed Abdalla Ayoub.

**Writing – original draft:** Fahd Saeed Alakbari.

**Writing – review & editing:** Fahd Saeed Alakbari, Mysara Eissa Mohyaldinn, Mohammed Abdalla Ayoub, Ali Samer Muhsan, Ibnelwaleed A. Hussein.

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
