## [Decision Letter · Decision Letter 0]

15 Feb 2021

PONE-D-20-39857

A Robust Fuzzy Logic-Based Model for Predicting  the Critical Total Drawdown in Sand Production in Oil and Gas Wells

PLOS ONE

Dear Dr. Mohyaldinn,

Thank you for submitting your manuscript to PLOS ONE. After careful consideration, we feel that it has merit but does not fully meet PLOS ONE’s publication criteria as it currently stands. Therefore, we invite you to submit a revised version of the manuscript that addresses the points raised during the review process.

I just received commendation and comments from one reviewer.Please address these comments.Please clearly highlight the novelty of your study.

We look forward to receiving your revised manuscript.

Kind regards,

Jianguo Wang, PhD

Academic Editor

PLOS ONE

Journal Requirements:

Reviewers' comments:

Reviewer's Responses to Questions

**Comments to the Author**

1. Is the manuscript technically sound, and do the data support the conclusions?

Reviewer #1: Yes

2. Has the statistical analysis been performed appropriately and rigorously? 

Reviewer #1: Yes

3. Have the authors made all data underlying the findings in their manuscript fully available?

Reviewer #1: Yes

4. Is the manuscript presented in an intelligible fashion and written in standard English?

Reviewer #1: Yes

5. Review Comments to the Author

Reviewer #1: This paper presented a robust fuzzy logic-based model to predict the CTD of sand production in oil and gas wells. The research is interesting and rather important. However, there exist a number of issues throughout the manuscript, especially the especially literature review, the proposed methodology, etc., that need to be addressed or clarified before it can be accepted for publication at the journal.

The main issues are as follows.

1)In Methodology Section, the fuzzy control model shall be specifically designed for the CTD prediction instead of the general architecture of fuzzy inference system (Fig.1). The value of the parameters in the model shall be defined clearly, for example, the membership function, the fuzzy rules, the ways of fuzzification and defuzzification, etc. Besides, the improvement and advantages of the proposed model shall be fully demonstrated to highlight the contribution of the manuscript.

2)Line 49, Artificial Intelligence (AI)? I did not see any AI description in the manuscript.

3)Line 58 - Line 106 , three techniques for predicting sand production are discussed. Are these techniques also used for predicting the CTD? Line 102 - 118 discussed the models to predict the CTD. The relationship of these two parts shall be explained.

4)Line 83, 84, etc, an FL approach? Or a FL approach.

5)Line 117, “the applicability of these models has significant limitations and their accuracy is questionable.” needs detailed explanation.

6)The literature review is missing. The current study of CTD predicting and the FL method shall be provided.

7)Line 181, “the sensitivity analysis or trend analysis is essential for this study” is confusing. The title of this section is ‘sensitivity analysis’, however the result analysis is based on the trend analysis.

6. PLOS authors have the option to publish the peer review history of their article (what does this mean?). If published, this will include your full peer review and any attached files.

Reviewer #1: No

---

## [Author Response · Author response to Decision Letter 0]

9 Mar 2021

Universiti Teknologi PETRONAS 32610 Seri Iskandar 

Perak Darul Ridzuan 

Malaysia 

+601169500177 

fahd_19001032@utp.edu.my

 November 23, 2020

Dear Prof. Joerg Heber

Editor-in-Chief

I am hereby resubmitting our revised manuscript based on the editor and reviewer comments. Please find enclosed the revised manuscript in two copies, highlighted copy and clean copy, along with a third file containing detailed responses to the reviewer. I would like to emphasize that all the requested changes have been addressed and indicated in the highlighted manuscript as well as the response to reviewers.

Best regards

---

## [Decision Letter · Decision Letter 1]

7 Apr 2021

A Robust Fuzzy Logic-Based Model for Predicting  the Critical Total Drawdown in Sand Production in Oil and Gas Wells

PONE-D-20-39857R1

Dear Dr. Mohyaldinn,

We’re pleased to inform you that your manuscript has been judged scientifically suitable for publication and will be formally accepted for publication once it meets all outstanding technical requirements.

Kind regards,

Jianguo Wang, PhD

Academic Editor

PLOS ONE

Additional Editor Comments (optional):

Reviewers' comments:

Reviewer's Responses to Questions

**Comments to the Author**

1. If the authors have adequately addressed your comments raised in a previous round of review and you feel that this manuscript is now acceptable for publication, you may indicate that here to bypass the “Comments to the Author” section, enter your conflict of interest statement in the “Confidential to Editor” section, and submit your "Accept" recommendation.

Reviewer #1: All comments have been addressed

2. Is the manuscript technically sound, and do the data support the conclusions?

Reviewer #1: Yes

3. Has the statistical analysis been performed appropriately and rigorously? 

Reviewer #1: Yes

4. Have the authors made all data underlying the findings in their manuscript fully available?

Reviewer #1: Yes

5. Is the manuscript presented in an intelligible fashion and written in standard English?

Reviewer #1: Yes

6. Review Comments to the Author

Reviewer #1: The authors addressed the reviewer's comments well, while improving the manuscript significantly.I have no further comments for this paper.

7. PLOS authors have the option to publish the peer review history of their article (what does this mean?). If published, this will include your full peer review and any attached files.

Reviewer #1: No

---

## [Editor Report · Acceptance letter]

12 Apr 2021

PONE-D-20-39857R1 

A robust fuzzy logic-based model for predicting the critical total drawdown in sand production in oil and gas wells 

Dear Dr. Mohyaldinn:

I'm pleased to inform you that your manuscript has been deemed suitable for publication in PLOS ONE. Congratulations! Your manuscript is now with our production department. 

Kind regards, 

on behalf of

Dr. Jianguo Wang 

Academic Editor

PLOS ONE